# ClimGen: Learning the Forcing-Response Relationship in Climate System

## Abstract

Solar Radiation Management (SRM) is emerging as a potential geoengineering strategy to address the climate change crisis, but its effective implementation requires an iterative and large ensemble of highly accurate and efficient climate projections. Traditional climate projections rely on executing computationally demanding and time-consuming numerical climate models. Recent advances in machine learning (ML) aim to enhance these approaches by emulating traditional methods. In this work, we propose a novel framework for directly learning the relationship between solar radiation flux at the top of the atmosphere and the corresponding surface temperature response. To evaluate the feasibility of this direct ML-based projection, we developed a benchmark dataset using an intermediate complexity model, incorporating a comprehensive suite of different forcing patterns and evaluation metrics to rigorously assess the ML model's performance. We introduce a Conditional Denoising Diffusion Probabilistic Model (cDDPM) for this task, which demonstrates superior performance in representing climate statistics under previously unseen forcing patterns. This approach provides a promising pathway for direct climate projections by accurately learning the forcing-response relationship, with a wide range of applications in climate change mitigation, emissions policy design, and SRM strategies.

## 1 Introduction

Reliable prediction of climate change response under external forcing and the uncertainty thereof has emerged as a central focus of climate research, especially in the context of Solar Radiation Management (SRM) as a potential geoengineering strategy to mitigate climate change. Initial efforts were limited to assessing the equilibrium climate sensitivity (ECS) that characterizes the long-term mean global temperature response in response to anthropogenic forcing from a doubling of the atmospheric $CO_2$ concentration (Roe (2009); Knutti et al. (2017)). More recent efforts have improved our understanding of the regional heterogeneity of the climate response from a linear perspective (Dong et al. (2019); Liu et al. (2022)). However, understanding and accurately predicting the full nonlinear climate response has remained challenging. This is further compounded by the chaotic nature of the Earth's climate system, its hierarchical structure, and the high dimensionality of the state space (Ghil & Lucarini (2020)). While running large ensemble Earth system models is the most reliable technique for climate prediction, it is compute-intensive. This puts a strong constraint on the number of climate scenarios and ensemble members that can be simulated and the number of parameters that can be perturbed for uncertainty quantification.

We have compiled data from a large suite of Green's function solar perturbation experiments that systematically probe the quantitative forcing-response relationship in an intermediate complexity climate model. Here, we use a denoising diffusion model to learn the emergent dynamic response function for global surface temperature conditioned on the applied solar forcing pattern. The temperature responses generated by the diffusion model also capture the inter-annual variability in the responses. Our approach aims to provide a cheap surrogate that can rapidly generate large ensembles of climate projections under various forcing scenarios while accurately capturing the internal variability in the climate response. Additionally, our trained model could be used for transfer learning by fine-tuning using minimal data to emulate the climate-forcing response in fully coupled Earth system models.

### 1.1 RELATED WORK

#### 1.1.1 LEARNING LINEAR FORCING-RESPONSE RELATIONSHIP IN CLIMATE

Our study falls into the broad category of climate emulators, specifically those that predict climate responses to external forcing. Many of the conventional methods used for climate emulation are based on simple pattern-scaling (Tebaldi & Arblaster (2014)), 1D impulse response function (Mac-Martin & Kravitz (2016)), and linear regression (Liu et al. (2022)). More importantly, these methods rely on the strong assumption that the climate response is sufficiently linear and time-invariant.

#### 1.1.2 ML FOR ATMOSPHERIC MODELING

Recently, deep learning-based autoregressive models have made significant progress in achieving efficient and accurate medium-range weather predictions (e.g., Keisler, 2022; Pathak et al., 2022; Bi et al., 2023; Lam et al., 2023). However, transferring this skill to long-term climate projections remains challenging due to error accumulation that emerges in forecast roll-outs beyond a two-week timescale. Efforts have been made to extend stable forecast horizons to up to 10 years under normal climate conditions (Weyn et al., 2021; Watt-Meyer et al., 2023; Guan et al., 2024; Cachay et al., 2024), but the generalizability of these autoregressive climate emulators to different climate forcing backgrounds has yet to be demonstrated.

An alternative to autoregressive climate emulation focuses on directly learning the forcing-response relationship. Recent efforts have employed machine learning techniques, such as random forests, Gaussian processes, and neural networks, to predict the full nonlinear climate response to potential future anthropogenic emission pathways (Watson-Parris et al., 2022). In this approach, the projected time variations in the global mean of anthropogenic greenhouse gas (GHG) concentrations across different scenarios serve as the input. However, these methods overlook the spatial forcing patterns associated with anthropogenic GHG emissions, limiting their ability to capture the full complexity of climate responses.

### 1.2 CONTRIBUTIONS

This study presents several novel contributions to climate modeling and projections. First, it utilizes a comprehensive range of forcing configurations generated from an intermediate-complexity climate model, as opposed to the limited scenarios typically derived from more complex models (Watson-Parris et al. (2022)). Second, it represents the first attempt to directly project the equilibrium state of the climate system from a forcing field using a generative AI technique, enabling a distributional learning of climate responses. This would allow for not only climate projection of mean but also extreme states. Our approach also learns the spatial relationship between the forcing and response patterns. Lastly, the model was validated with an additional test case using a realistic climate model with $CO_2$ forcing, where it successfully produced a consistent global mean temperature distribution, demonstrating its potential applicability to more complex and realistic climate scenarios.

## 2 CLIMATE FORCING AND RESPONSE DATASET

The data used in our study is generated using a widely used intermediate complexity climate model called Planet Simulator (PlaSim) (Fraedrich et al. (2005)). The dynamical core of PlaSim is a simplified global circulation model (GCM) of the atmosphere with parameterization schemes for modeling physical processes such as diffusion, radiative processes, moist processes including cloud formation and precipitation, and dry convective adjustment. It also includes linearized representations of important Earth system components such as a slab ocean with sea ice and a land surface with biosphere. The grid resolution for the simulations is chosen to be $5.6° \times 5.6°$.

To probe the patterned forcing-response relationship in the climate system, a large number of Green's function experiments, in the vein of Hassanzadeh & Kuang (2016); Liu et al. (2022), are conducted by perturbing the incoming solar radiation at the top of the atmosphere (TOA) over localized patches ($16 \times 16$) distributed around the globe as shown in Fig 1. Note that applied forcing is seasonally varying as shown in the schematic such that the annual mean of the forcing is roughly the same across all the patches irrespective of the location. We use six different forcing levels

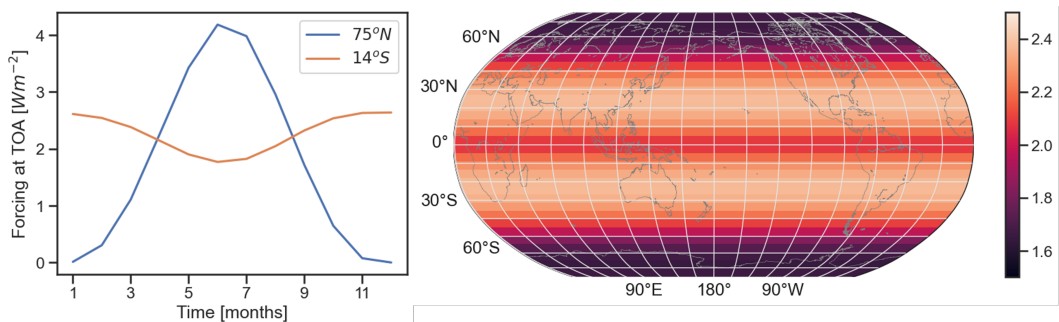

Figure 1: Configuration of the Green's function experiments: The right panel illustrates the 256 local patches distributed around the globe on top of uniform $2Wm^{-2}$ TOA forcing field, while the left panel displays the TOA forcing over two different latitudinal patches over the course of a year.

($\pm 15\ Wm^{-2}$, $\pm 30\ Wm^{-2}$, and $\pm 60\ Wm^{-2}$). Using multiple negative and positive forcing levels allows us to quantify the nonlinearities in the surface temperature response systematically. A preindustrial control simulation is carried out for 150 years and the data from the last 100 years is used as the baseline reference (unforced) state. The control data also provides a robust estimate of the internal variability in the climate. The forced simulations were started from the $100^{th}$ year of the control simulation and ran for another 60 years. The data from the last 20 years of the forced runs is used for training and testing the cDDPM to ensure sufficient equilibration of the climate. The dataset also includes uniformly forced runs at $\pm 2\ Wm^{-2}$, $\pm 4\ Wm^{-2}$, and $\pm 8\ Wm^{-2}$ (see for example Figure 1).

The diffusion model was trained on the Green's function runs data and four of the uniformly forced runs– $\pm 4\ Wm^{-2}$ and $\pm 8\ Wm^{-2}$. The uniformly forced $\pm 2\ Wm^{-2}$ cases are set aside for testing.

## 3 CONDITIONAL DIFFUSION MODEL FOR CLIMATE PROJECTION

We employ a conditional denoising diffusion probabilistic model (cDDPM) to generate an ensemble of climate projections conditioned on the applied forcing pattern.

### 3.1 DENOISING DIFFUSION PROBABILISTIC MODEL

The denoising diffusion probabilistic model introduced in Ho et al. (2020) belongs to the class of latent variable models; it involves a forward process that gradually adds noise to the data and a reverse process that learns to retrieve the original data through systematic denoising. The forward diffusion is a Markov process with a fixed number of steps $n$, during which Gaussian noise is iteratively added to the input $\mathbf{x}_0$. At the $i^{th}$ step, $\mathbf{x}_i$ is obtained by noising the preceding iterate $\mathbf{x}_{i-1}$ based on a prescribed variance schedule ($\beta_i \in (0, 1)$) such that,

$$q(\mathbf{x}_i|\mathbf{x}_{i-1}) = \mathcal{N}(\mathbf{x}_i; \sqrt{1-\beta_i}\mathbf{x}_{i-1}, \beta_i\mathbf{I}). \tag{1}$$

Equivalently, $\mathbf{x}_i$ can be directly obtained from $\mathbf{x}_0$ as

$$q(\mathbf{x}_i|\mathbf{x}_0) = \mathcal{N}(\mathbf{x}_i; \sqrt{1-\bar{\alpha}_i}\mathbf{x}_0, \bar{\alpha}_i\mathbf{I}), \tag{2}$$

where $\bar{\alpha}_i = \prod_{s=1}^{i} \alpha_s$ and $\alpha_s = 1 - \beta_s$.

The reverse process is also a Markov chain such that $\mathbf{x}_{i-1}$ at the $i^{th}$ denoising step is obtained as

$$p_\theta(\mathbf{x}_{i-1}|\mathbf{x}_i) = \mathcal{N}(\mathbf{x}_{i-1}; \mu_\theta(\mathbf{x}_i, i), \Sigma_\theta(\mathbf{x}_i, i)). \tag{3}$$

Here, $\mu_\theta(\mathbf{x}_i, k)$ is the learned mean from a trained neural network and $\Sigma_\theta(\mathbf{x}_i, i)$ is the covariance matrix. For simplicity, the reverse process employs a fixed covariance matrix ($\sigma_i^2\mathbf{I}$ with $\sigma_i^2 = \beta_i$) that mirrors the forward diffusion. The neural net is a function approximator that predicts the noise $\epsilon$ from $x_i$, $\epsilon_\theta(\mathbf{x}_i, i)$ where the subscript $\theta$ represents the trained network parameters. The mean

$\mu_\theta(\mathbf{x}_i, i)$ is then computed as

$$\mu_\theta(x_i, i) = \frac{1}{\sqrt{\alpha_i}} \left( \mathbf{x}_i - \frac{1 - \alpha_i}{\sqrt{1 - \bar{\alpha}_i}} \epsilon_\theta(\mathbf{x}_i, i) \right). \tag{4}$$

The parameters $\theta$ are obtained by minimizing the loss $\mathcal{L}_\theta = \mathbb{E}_{i, \mathbf{x}_0, \epsilon} \left( \|\epsilon - \epsilon_\theta(\mathbf{x}_i, i)\|^2 \right)$.

## 3.2 CLASSIFIER-FREE CONDITIONING

To generate ensembles of the climate response for a given solar forcing pattern, we added classifier-free conditioning to the DDPM (Ho & Salimans (2022)). This is achieved by modifying (3) to account for the condition $\mathbf{c}$ and is given by,

$$p_\theta(\mathbf{x}_{i-1}|\mathbf{x}_i, \mathbf{c}) = \mathcal{N}(\mathbf{x}_{i-1}; \mu_\theta(\mathbf{x}_i, i, \mathbf{c}), \Sigma_\theta(\mathbf{x}_i, i)). \tag{5}$$

Here, the condition $\mathbf{c}$ is an embedding generated by a fully connected neural network for a given forcing pattern $\mathbf{F}$ as shown in Figure 3. Similarly, the loss function to be optimized becomes $\mathcal{L}_\theta = \mathbb{E}_{i, \mathbf{x}_0, \epsilon} \left( \|\epsilon - \epsilon_\theta(\mathbf{x}_i, i, \mathbf{c})\|^2 \right)$ and the mean $\mu_\theta(x_i, i)$ is computed as

$$\mu_\theta(x_i, i) = \frac{1}{\sqrt{\alpha_i}} \left( \mathbf{x}_i - \frac{1 - \alpha_i}{\sqrt{1 - \bar{\alpha}_i}} \epsilon_\theta(\mathbf{x}_i, i, \mathbf{c}) \right). \tag{6}$$

## 3.3 U-NET BACKBONE

We use a standard U-Net architecture composed primarily of five ResNet blocks, as the backbone for our cDDPM. The details of the U-Net backbone used in our study, along with the time and context embedding blocks are shown in Figures 2 and 3. We also tested several variants of this architecture by varying the location of the context and time embedding and by introducing a self-attention block at the bottleneck of the U-Net; the results comparing the performance of these variants are described in section 5.

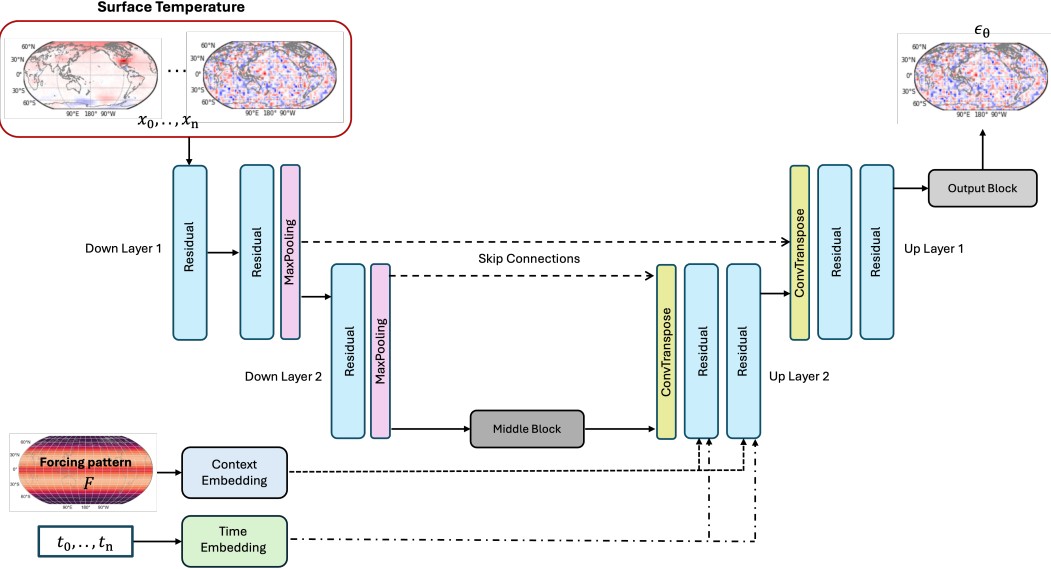

Figure 2: Schematic of the U-Net

### 3.3.1 CONTEXT EMBEDDING

For conditioning, we utilize a natural reduced-order representation of the forcing pattern that reflects the resolution at which the forcing-response relationship is represented in the Green's function experiments described in Section 2. The full forcing field with a resolution of $32 \times 64$ is reduced to a $16 \times 16$ matrix with each entry representing the annual mean of the applied solar perturbation over

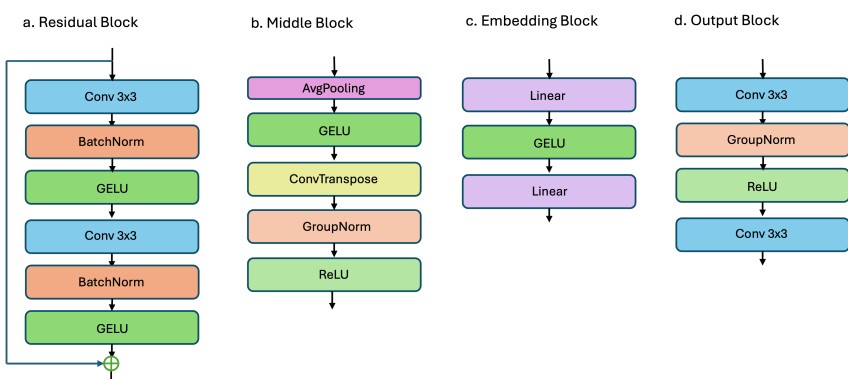

Figure 3: Detailed structure of the component blocks of the U-Net backbone.

the Green's function patches shown in the Figure 1. Direct conditioning on the forcing field at full resolution ($32 \times 64$) was intractable and resulted in poor prediction of the temperature response.

The context embedding is achieved via an embedding block that consists of two linear transforms and a GELU activation as shown in Figure 3c.

### 3.3.2 SELF-ATTENTION

We also tested a variant of the basic architecture that includes a simple self-attention block at the bottleneck of the U-Net; specifically at the beginning of the middle block shown in Figure 3.

## 4 RESULTS

Here we evaluate the performance of the trained cDDPM on two independent test cases: uniform $2Wm^{-2}$ and -2 $Wm^{-2}$ forcing perturbations. Figure 4 presents 12 generated samples for each forcing scenario. It is evident that while the large-scale patterns across the samples are quite similar, each sample retains its own inter-annual variability under the same forcing.

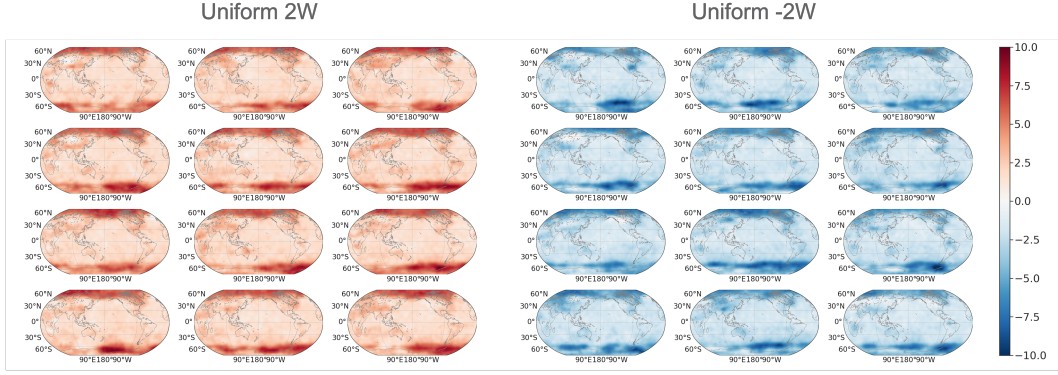

Figure 4: Ensemble of cDDPM-generated surface temperature responses to uniform $\pm 2Wm^{-2}$ forcing perturbations.

To better assess the performance of cDDPM and the quality of the generated samples, we employed 3 temperature metrics introduced in Kravitz et al. (2017): the global mean surface temperature ($T_0$), the interhemispheric surface temperature gradient ($T_1$), and the equator-to-pole surface temperature gradient ($T_2$). These metrics are defined as follows:

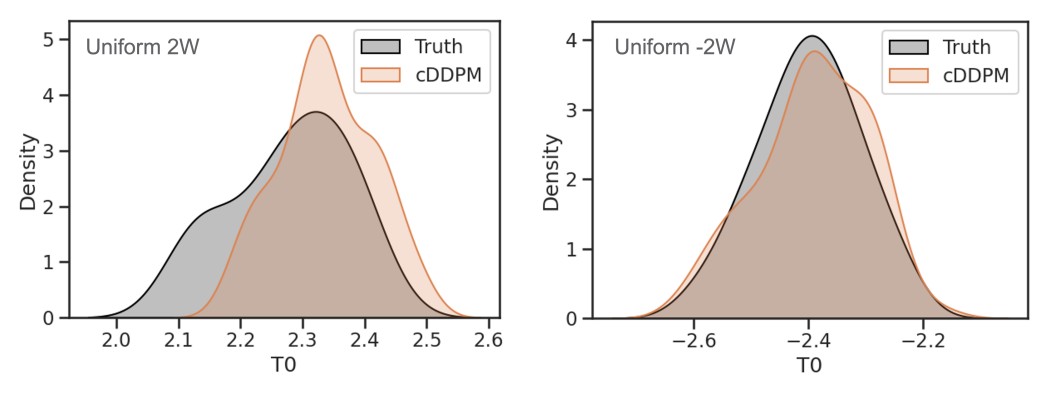

Figure 5: Comparison of the cDDPM-generated with the true distribution of global mean temperature changes $[K]$ under uniform $\pm 2Wm^{-2}$ forcing perturbations.

$$T_0 = \frac{1}{A} \int_{-\pi/2}^{\pi/2} T(\psi) dA \tag{7}$$

$$T_1 = \frac{1}{A} \int_{-\pi/2}^{\pi/2} T(\psi) \sin(\psi) dA \tag{8}$$

$$T_2 = \frac{1}{A} \int_{-\pi/2}^{\pi/2} T(\psi) \frac{1}{2}(3\sin^2(\psi) - 1) dA \tag{9}$$

where $dA = 2\pi R^2 \cos(\psi)d\psi$ is the area of a latitudinal band, and $A = 2\pi R^2 \int_{\psi=-\pi/2}^{\psi=\pi/2} dA$ is the total surface area of the Earth.

We compare the distributions of the cDDPM-generated and the true changes in these temperature metrics relative to the reference climate (without forcing perturbation) under the 2 independent forcing scenarios. Figure 5 compares cDDPM-generated and true distributions of global mean temperature changes under the two forcing scenarios. The similarity in both the peak and width of the generated and the true distributions indicates that the cDDPM can produce samples with representative probability distributions for global mean temperature changes in response to forcing perturbations.

Notably, the responses to positive and negative forcing perturbations of the same magnitude are 2.27K and -2.40K, respectively, and are not symmetric around zero. The differing peak locations and distribution shapes further emphasize the nonlinear nature of the climate system, demonstrating that the cDDPM effectively captures this inherent nonlinearity.

We also compare the joint and marginal distributions of $T_1$ and $T_2$ in Fig 6. In the uniform +2 $Wm^{-2}$ scenario, the spread of both temperature metrics aligns well between the cDDPM-generated and true distributions, although the cDDPM underestimates the peak of $T_1$. In the uniform -2 $Wm^{-2}$ scenario, the cDDPM generates a representative width, but the peak location of both $T_1$ and $T_2$ shows a more noticeable underestimation. Despite these differences, the cDDPM is still capable of generating representative samples under the given forcing perturbations..

Finally, we assess the performance of the cDDPM by validating its equilibrium climate sensitivity (ECS), defined as the global mean surface temperature response to a doubling of atmospheric $CO_2$. The necessary forcing field was obtained from the Community Earth System Model (CESM). Figure 7 (left) shows the original forcing perturbation field alongside the coarsened version used as the cDDPM context input, while Figure 7 (right) presents the cDDPM-generated $T_0$ distribution.

Although there is no definitive ground truth for ECS, we compared the cDDPM's response with estimates from state-of-the-art climate models, specifically the CMIP5 and CMIP6 simulations. The model range of ECS for CMIP5 is 2.1 to 4.7K, and for CMIP6, it is 1.8 to 5.6K (Schlund et al., 2020; Meehl et al., 2020). The ECS range generated by the cDDPM falls between approximately 2.7 to

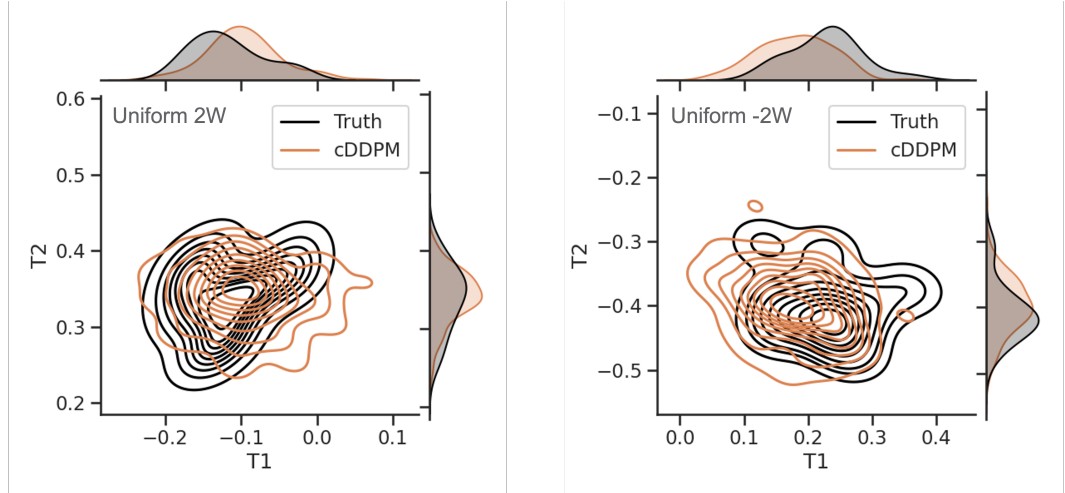

Figure 6: Comparison of the cDDPM-generated and true joint and marginal distributions of $T_1$ and $T_2$ under uniform $\pm 2Wm^{-2}$ forcing perturbations

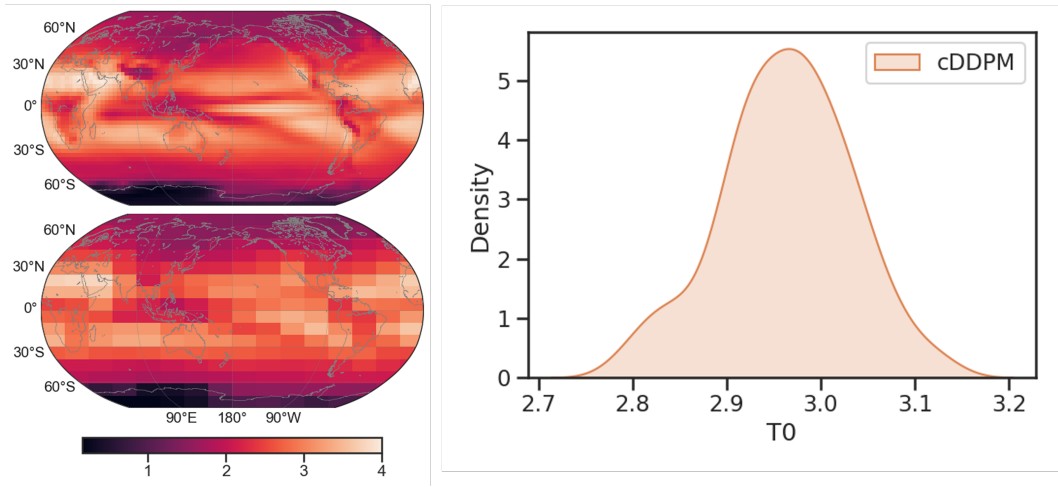

Figure 7: Left panel shows the forcing field from a CESM doubling $CO_2$ warming scenario. We coarsen the resolution from the original size of 48x96 (top) to the 16x16 context input size (bottom) for cDDPM. Right panel shows the distribution of an ensemble of the cDDPM-generated mean surface temperature response.

3.2K, well within the ranges provided by these advanced climate models. This comparison suggests that, despite being trained on data from an intermediate-complexity climate model, the cDDPM is capable of producing realistic global mean temperature responses consistent with more complex and computationally intensive climate models.

## 5   ABLATION STUDIES

To identify the best-performing model, we experimented with several variants of the cDDPM by modifying the backbone U-Net architecture. These variants differed in the location of the context embedding within the U-Net and the inclusion of a self-attention block at the bottleneck.

Table 1 outlines the architectural differences among the variants and compares their performance using a comprehensive set of error metrics: RMSE (root mean squared error), NRMSE (normalized root mean squared error), MAE (mean absolute error), and ACC (anomaly correlation coefficient).

| | $cntx_{all}$ | $attn\_cntx_{all}$ | $cntx_{up}$ | $attn\_cntx_{up}$ | $cntx_{dn}$ | $attn\_cntx_{dn}$ |
|---|---|---|---|---|---|---|
| Self-attention block | No | Yes | No | Yes | No | Yes |
| Downward-layer context embedding | Yes | Yes | No | No | Yes | Yes |
| Upward-layer context embedding | Yes | Yes | Yes | Yes | No | No |
| RMSE | 0.500 | 0.428 | 0.418 | 0.491 | **0.336** | 0.347 |
| NRMSE | 0.00173 | 0.00148 | 0.00145 | 0.00170 | **0.00116** | 0.00120 |
| MAE | 0.367 | 0.305 | 0.315 | 0.382 | **0.229** | 0.234 |
| ACC | 0.980 | 0.985 | 0.988 | 0.986 | **0.991** | 0.990 |
| Bias (2W) | 0.100 | **0.0242** | 0.232 | 0.180 | 0.0604 | 0.0652 |
| Bias (-2W) | 0.0203 | **-0.000822** | -0.0608 | -0.296 | 0.00797 | -0.0446 |
| RMSE (linear) | 0.365 | 0.303 | 0.329 | 0.394 | **0.245** | 0.265 |
| RMSE (nonlinear) | 0.347 | 0.302 | 0.259 | 0.294 | 0.232 | **0.226** |
| ACC (linear) | 0.989 | 0.992 | 0.993 | 0.992 | **0.995** | 0.994 |
| ACC (nonlinear) | 0.348 | 0.378 | 0.505 | 0.572 | 0.589 | **0.604** |

Table 1: Performance scoreboard for independent test cases comparing cDDPM variants.

Additionally, biases for the positive and negative forcing cases were analyzed, and we further assessed RMSE and ACC for both linear and nonlinear components of the temperature response as defined in Lu et al. (2020). Overall, the $cntx_{dn}$ variant demonstrated the best performance across most error metrics and consistently ranked as the second best even when it did not achieve the top score.

## 6    CONCLUSIONS AND FUTURE WORK

In this study, we developed a conditional denoising diffusion model with a U-Net backbone to learn the relationship between forcing perturbations and climate system responses. This proof-of-concept study with a minimalist cDDPM demonstrates the potential for training a direct distributional projection of global temperature responses based on the context of a forcing perturbation field as input. Notably, this work represents the first attempt to train a machine learning model for climate projection using a comprehesive set of forcing configurations generated by an intermediate-complexity climate model. The generated samples were validated against independent test cases using various temperature metrics, showing that the cDDPM is capable of producing samples that are representative of both the intermediate-complexity model and the even more realistic and computationally intensive CMIP models.

This endeavor will prove invaluable for a wide range of applications, including climate change mitigation efforts, the development of effective emissions policy designs, and the exploration of SRM strategies. By providing a more accurate and efficient means of projecting climate responses to various forcing scenarios, this approach can inform policymakers and scientists in crafting data-driven strategies to reduce greenhouse gas emissions, implement adaptation measures, and assess the potential risks and benefits of geoengineering techniques.

Future work will focus on exploring alternative backbone architectures to enhance performance further. We also aim to extend the model's capabilities to predict precipitation, which poses an even greater challenge due to its larger nonlinearity compared to temperature. Moreover, one potential challenge in directly projecting climate responses from forcing perturbations is the limited training data available in terms of the variety of forcing configurations. However, we suggest that the dataset used in this study could serve as a valuable resource for pre-training foundational models like the cDDPM we developed, which could then be fine-tuned using outputs from more realistic climate models.

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
