# OpenReview forum: "ClimGen: Learning the Forcing-Response Relationship in Climate System"
_ICLR.cc/2025/Conference — ICLR 2025 Conference Withdrawn Submission_

### Official Review · Reviewer_1KWw · 2024-10-18

**Soundness:** 2
**Presentation:** 4
**Contribution:** 2
**Rating:** 6
**Confidence:** 4

**Summary:**

The study uses a combination of intermediate climate model simulations and diffusion-based framework to understand the forced-response of the climate system using a green's function approach. The problem could potentially contribute to addressing solar radiation management strategies aimed at inducing global cooling through modulation of incoming solar radiation.

Perturbing an intermediate complexity climate model (with multiple parameterizations) by varying forcing across 256 patches provides training data for the conditional diffusion model probabilistic model. The diffusion model is trained on multiple time-varying forcing scenarios and is tested on more comprehensive scenarious with spatial variations.

The authors claim and the results somewhat suggest that training diffusion models on intermediate model data trasfers well to predict climate response in more complex climate models.

The training data generated for the study can also be used by future studies to develop ML emulators for climate models for varying climate scenarios.

**Strengths:**

1) The problem of SRM is topical and the application of ML/AI in addressing the problem is quite relevant. The application of ML/AI in SRM is quite limited. Moreover, large uncertainties in climate models prohibit decision making

2) The experimental setup of the study is sound and the training data chosen is of good quality. The study claims that the forcing scenarios in the study are more comprehensive that those considered in the previous studies (for ex., the ClimateBench)

3) The mathematical foundation of the problem is sound: Understanding the net response of a system to comprehensive forcing scenarios by treating it as a linear combination of multiple green's function responses

4) As per my understanding the methodology is described in good detail, however I have some reservations regarding the experimental setup and its robustness and some modeling choices used in the study.(see below)

5) The paper is quite well-written. The machine learning techniques used are clearly defined. The schematic provides a clear description of the architecture. The introduction is well-written and the results, in many places, support the conclusions. However, I do believe some conclusions have been extrapolated too much without proper evidence (see below)

6) The work illustrates a creative application of a blend of ML architectures and ensemble modeling. My belief is that the data set wilbe of value to the climate community and the experimental setup might inspire more studies focused on the development of climate emulators for climate prediction.

**Weaknesses:**

While I elaborate on this in the "Questions" section below, here are some of the aspect which I find to be weak:

1) The study does not acknowledge any weak links in their findings. The predictive ensemble does not seem to capture the distributions as well as the text appears to claim and it is not acknowledged in the paper

2) The paper reduces the climate system to a much lower dimensional problem that it should be. Solar radiation does provide a key forcing on the earth system, but the study aims to create a direct relationship between the solar radiation and the surface temperature, ignoring the complex dynamics

3) Testing the model on the +/-2 W/m2 forcing is not a good choice to demonstrate the effectiveness of diffusion models in predicting surface response. Especially as a lot of uncertainty in climate models enters through parameterization. At 5.6 degree resolution, an intermediate complexity model parameterizes a lot of processes. Both factors combined, this makes me question a bit the quality of the dataset and the validity of the results.

**Questions:**

1) I struggled a bit to understand the details of Section 2 a bit. I think the Green's function experiment setup description could use some more details. For instance, I don;t fully understand the right panel in Figure 1. Te caption says its a uniform field with 16x16 patches, but the field is clearly not uniform. Also, if the forcing for each patch is time varying only over the whole year, there is no interannual feedback  assumed while defining the forcing. Also, is the annual variation for all patches along the same latitude circle identical?

2) A major issue I have with the study is their choice of 2 W/m2 for testing the ML model performance. I believe this is not a god choice and would like to understand the reason behind this. Why was the model not tested on (better) 4 W/m2 or (best) 8 W/m2 which would be a more rigorous test of the model? 2 W/m2 is quite close to the control and does not fully represent the future warming scenarios.

3) L176: I think you mean Figure 2 instead

4) Clarification: Figure 5 shows surface temperature and not air temperature, right?

5) Figure 5: I think the conclusion is a bit misleading. The +2W distribution is not relaly identical. It has a weaker spread and the predictions overestimate the clustering around mean global temperature rise. Also, it would be better to show the both the true and predicted distribution mean to support the statement on L308-309

6) L312 - I think you mean overestimates and not underestimates?

7) L323 - is this the range for the 2.6 W/m2 forcing scenario in CMIP?

8) L365 - Good! But it should be acknowledged why the emsemble dispersion in predicting ECS is much weaker than CMIP models and how it can be fixed.

9) Table 1: Seems like the most prominent factor in determining model performance is the upward layer context embedding. Models with no upward layer embedding perform better. Why does this embedding degrade performance?

---

### Official Review · Reviewer_kXbH · 2024-11-03

**Soundness:** 1
**Presentation:** 2
**Contribution:** 3
**Rating:** 3
**Confidence:** 4

**Summary:**

The authors outline an approach to learn the response relationship between the top of the atmosphere solar radiation flux and surface temperature, effectively learning to act as an emulator of a computationally expensive climate model. This has traditionally been a difficult problem for data driven approaches due to the lack of high quality data. The authors tackle this by generating a novel benchmark dataset using an intermediate complexity general circulation model and then training a conditional diffusion probabilistic model to act as the emulator. The motivation for this work is strongly based on the wide ranging potential impacts.

**Strengths:**

The generation of a novel dataset of forced climate simulations is a key strength of this paper. Training a probabilistic model offers the ability to explore extreme responses to the forcing fields.

**Weaknesses:**

- No comparison to baselines is carried out. Where appropriate, the author(s) could implement the baseline models from ClimateBench.
- There is a l ack of validation of both the intermediate complexity model and the trained data driven model against the fully fledged climate model. Validation against existing fully fledged climate model results should be added (ClimateBench).
- There is a lack of consideration of tipping points in the climate system, this reviewer understands the difficulty of including these but a comment addressing these concerns should be added.
- Details on the computational requirements (runtimes, core counts etc.) to run the intermediate complexity model are missing. This is necessary to understand the drivers to train the data driven model. If the intermediate model is computational cheap to simulation (which I suspect it is) then training an emulator of it is a moot point.
- Considerations of the biases of the intermediate complexity model are missing and should be included to fully understand the outputs of the data driven model.   Pertinent to the results presented here, the temperature biases of the intermediate complexity model over the past climate should be presented. Ideally these biases should be address when building the emulator.

**Questions:**

- Add an appendix containing greater detail on the generated data   and the biases associated with the intermediate complexity model.
- Lean into the distributional component of the learned model and try to derive insights on the extremes
-  Add a comparison against an existing data driven baseline approach to figure 7
- Add a data availability statement
- Can the authors provide further details on the performance of the approach when the full forcing field is used for conditioning?   Why have they chosen to reduce the dimension to a 16x16 array? Ideally one should reduce original latitude and longitude dimensions by a common factor. Are there any technical limitations that led to the 16x16 reduction choice? Evaluating model performance against forcing field dimension should showcase the benefit of using the 16 X16 array.
- It is noted that the validation results in section 4 are for a +/- 2 W m^-2, which given the make-up of the training dataset acts as an interpolation task. Have the authors tried a scenario which would push the model outside the training corpus?
- The authors should be explicit as to where the ground truth distributions in Figure 5 and 6 come from - the intermediate complexity climate model?
- In Figure 7 (right panel) the authors should also provide an estimate of the ECS from the intermediate complexity model.

---

### Official Review · Reviewer_AWGc · 2024-11-03

**Soundness:** 1
**Presentation:** 3
**Contribution:** 1
**Rating:** 3
**Confidence:** 5

**Summary:**

The paper uses a conditional denoising diffusion model to model the climate's response to solar radiation forcing. Using an intermediate-complexity climate model, the authors create a benchmark dataset and directly learn the relationship between solar forcing and surface temperature. This approach enables efficient, probabilistic climate projections and has potential applications in geoengineering.

**Strengths:**

Building climate emulators is important for advancing climate science, as they offer a faster means of simulating and understanding climate dynamics under varying scenarios. Machine learning has significant potential in this area. The application of a conditional denoising diffusion probabilistic model in this paper brings a fresh and promising approach. The benchmark dataset provides a systematic suite of forcing configurations, offering a good resource for researchers aiming to validate and compare new emulation techniques.

**Weaknesses:**

- Although the training data is diverse, including both Green's function experiments and uniformly forced scenarios across several forcing levels, the test dataset is restricted to only the lowest magnitude of uniformly forced experiments. This limited test scope raises concerns about the model’s ability to generalize effectively, as it hasn’t been evaluated on higher or more varied levels and different types of forcing. The test set would be benefit from including a broader range of forcing. A more comprehensive test set would ultimately support a clearer understanding of the model's generalization capabilities, which is extremely important for climate study, where we don't have any observations of future climate to validate our current approach.

- The results presented do not fully support the claims made in the paper. Even with the limited test cases, performance appears inconsistent. While the $T_0$ distribution generated by cDDPM for the uniform $-2 Wm^{-2}$ case aligns reasonably with the reference distribution (Figure 5, right), the distribution for the uniform $+2 Wm^{-2}$ case is noticeably inaccurate (Figure 5, left). Furthermore, the joint and marginal distributions of $T_1$ and $T_2$ generated by cDDPM differ significantly from the reference distributions, as shown in Figure 6. In the uniform $+2 Wm^{-2}$ scenario, the predicted spread of $T_1$ is larger than the reference spread, while the spread of $T_2$ is smaller. This contradicts the authors' assertion that the spreads “align well.” In the uniform $-2 Wm^{-2}$ scenario, although the spread of $T_2$ is acceptable, the overall distributions remain noticeably different from the reference, with a substantial shift in the distribution of $T_1$. Under these conditions, describing the spread as “representative” is questionable, as it fails to reflect the true distribution shapes and locations. Given these inconsistencies, it is difficult to justify the conclusions stated in lines 303-305 and lines 313-315 that the cDDPM "can produce samples with representative probability distributions" and "captures the spread well." To substantiate these claims, the authors would need to demonstrate stronger alignment across all distributions or consider alternative ways to improve model fidelity, and should demonstrate the result on more test cases, as mentioned above.


- Unfounded claim of nonlinearity capture. The authors claim that the cDDPM effectively captures the inherent nonlinearity in the relationship between climate forcing and response. However, this assertion lacks sufficient evidence. While the response to positive and negative forcing perturbations of the same magnitude does differ and is not symmetric around zero, it is unclear whether these differences are statistically significant. Without further analysis, it is premature to conclude that the model captures true nonlinearity in the climate system. The different peak locations and distribution shapes produced by the cDDPM can indeed suggest a form of nonlinearity, but this reflects only the internal structure of the model itself rather than the actual nonlinear dynamics of the climate system. In fact, the true distributions of uniform $\pm 2Wm^{-2} $ exhibit very different shapes. However, the distributions generated by the cDDPM show a similar pattern (Figure 5), suggesting that the model is not yet capturing the climate nonlinearity.

- Lack of physical interpretability/constraint. The use of a conditional denoising diffusion probabilistic model has been seen in many ML for weather and climate studies. The core model itself functions as a 2D image conditional generation process without physical constraints. This would limit the generalization ability, especially extrapolation on physical parameters.

- The choice to coarsen the CO$_2$ forcing field, while computationally convenient, introduces localized inaccuracies that may limit the model's ability to make precise predictions. For instance, in Figure 7, the CO$_2$ concentration along the east coast of South America is significantly altered after coarsening, resulting in discrepancies that could undermine the model's effectiveness for localized predictions and, consequently, limit its use in decision-making scenarios that require fine-scale accuracy. The justification for this coarsening—conditioning on a $16 \times 16$ grid because higher resolutions are "intractable and result in poor prediction"—seems questionable. This limitation could reflect a model configuration or hyperparameter-tuning issue rather than a fundamental constraint, suggesting that with better optimization, conditioning on higher resolutions might be feasible. As it stands, the decision to coarsen appears more like a compromise than a scientifically rigorous choice, potentially impacting the credibility and applicability of the results.

- The ablation studies conducted primarily focus on variations within the cDDPM architecture, such as different configurations of the U-Net backbone. However, these structural adjustments do not contribute meaningfully to validating the central claims of the paper. The lack of comparisons with baseline models commonly used in climate emulation—such as Gaussian Processes—limits the ability to evaluate the unique advantages or performance gains of cDDPM in this context. Including standard baseline models would provide a clearer reference point and demonstrate the specific strengths of cDDPM in capturing climate forcing-response relationships. Without these baselines, it is challenging to isolate and assess the impact of using cDDPM over more traditional or simpler emulation methods.

- The intermediate-complexity model used to generate data, if it is indeed the cited 2005 version, appears to lack a dynamic representation of the ocean and sea ice and does not include a carbon cycle model. This omission means that it overlooks significant Earth system feedbacks, which are essential for accurately predicting temperature responses, especially in future climate scenarios. This dataset is not the best for climate emulator study now.

- The paper’s generalization to more complex Earth System Models (ESMs) is also limited; it only demonstrates that the spread of cDDPM-predicted outcomes falls within the range of CMIP5 and CMIP6 outputs. However, it does not compare any other meaningful metrics that could provide a deeper validation of model performance, nor does it visualize the distributions to offer a more comprehensive comparison. This approach limits the strength of the generalization claims, as it fails to fully demonstrate cDDPM’s alignment with the complexity of high-fidelity ESM outputs.

**Questions:**

Please see the suggestions listed in the Weaknesses section. Given the importance and complexity of this topic, the current work could benefit from improvements in several areas. The dataset used is limited in scope and does not capture the full range of climate variability needed for robust climate emulation. Additionally, the methodology could be enhanced by incorporating physical constraints or more refined conditioning strategies to improve model accuracy and relevance. The results, while promising, are preliminary and do not yet fully support the authors' claims. Significant work remains to be done, and the authors are encouraged to enhance this research from the perspectives of data, modeling, and methods to reach a more comprehensive and reliable outcome.

---

### Official Review · Reviewer_qw46 · 2024-11-04

**Soundness:** 2
**Presentation:** 2
**Contribution:** 1
**Rating:** 3
**Confidence:** 4

**Summary:**

This paper proposes emulating the relationship between incoming solar radiation and resulting global surface temperatures using a conditional diffusion model (cDDPM). This is motivated by the high computational demands of running climate simulations with physics-based models, which present a bottleneck for more comprehensive climate projections. To do so, the authors generate training data for the emulator based on solar perturbation experiments using a coarse, intermediate-complexity climate model. The results suggest that the cDDPM does a reasonable job of projecting global mean temperature fields.

**Strengths:**

-  Developing fast yet reliable climate emulation methods that could make climate projections across different emission scenarios more accessible is a vital challenge. A probabilistic approach, as explored in this paper, is crucial in light of the large uncertainties associated with climate projections.
- Alternative benchmark datasets to ClimateBench is an important direction.

**Weaknesses:**

1. Implementation details are very scarce:
- No hyperparameter information is given (e.g. architectural details).
- No optimization details are given.
- No specifics about the diffusion model training (e.g. which noise schedule) and sampling (e.g. how many sampling steps) are given. It's also unclear whether classifier-free guidance or explicit, direct conditioning of the denoiser network was used for conditioning on the input forcings.
- It's unclear if/how a validation set was used and how the final model was chosen.

2. No baselines are included. Simple baselines are omitted:
- When proposing a new benchmark dataset it is vital, in my opinion, to include multiple baselines. In particular, it's important to include simple baselines. Directly jumping to using a diffusion model, which is a fairly advanced ML method, is questionable.
- It has been shown for the highly related ClimateBench dataset that extremely simple, non-neural ML approaches can excel at the problem, especially for emulating surface temperature responses (which is the only variable explored in this paper). For example, see the linear regression/pattern-scaling approach discussed in [1]. For probabilistic methods, you may want to compare with Gaussian Process-based methods as discussed in ClimateBench and ClimateSet [2].
- The abstract mentions that the proposed cDDPM *"demonstrates superior performance"* but without baselines such claims of superiority lack empirical support.

3. Related work is missing. The paper is insufficiently contextualized to very related papers. Some claims are imprecise.
- NeuralGCM [3] is an "autoregressive climate emulator" that has shown very promising generalizability to realistic climate forcings, in contrast to what the authors suggest in the related work section.
- ClimateSet [2] should be discussed given that it's a more recent, comprehensive benchmark dataset for learning "forcing-response
relationships" than ClimateBench.
- I am missing a more thorough comparison to ClimateBench and ClimateSet. The only discussed difference in this paper is that *"However, these methods overlook the spatial forcing patterns associated with anthropogenic GHG emissions (...)"*. Do you mean that the input forcings in ClimateBench/Set are non-spatial? I don't think this is true.
- Can you discuss how this paper relates to [4]? It seems quite relevant to this work.

4. The significance of the dataset and problem setup are unclear to me (perhaps due to insufficient contextualization to related work, see above):
- The abstract claims that a *"benchmark dataset"* was developed but the paper includes insufficient information about basic things such as its creation, documentation, accessibility, adherence to FAIR principles, code availability, etc.
- The climate model used to generate the training data is simplified. E.g. i) it's very coarse (32x64; which is coarser than even CMIP6 climate models and thus ClimateBench/Set); ii) the climate model itself is based on simplified modeling assumptions such as a slab ocean. I would have liked to see a discussion of the practical relevance of this setup and why more realistic training data simulators are not used (e.g. is it infeasible to run them with enough forcing scenarios; Why not simply use CMIP6 outputs?).
- The selling point of cheap, ML-based emulators is that they can speed up highly complex realistic climate models, as discussed by the paper. However, the paper could not convince me why emulating a simplified climate model is a good use case too.

5. The ML component of the paper has limited novelty/originality. The proposed method is a direct application of a standard conditional diffusion model.

6. Experiments need to be expanded:
- No runtime information is included. It's important to understand the runtime complexity of both the climate model used to generate the training data as well as the cDDPM, especially since a diffusion model is a rather runtime-expensive ML method. Including the aforementioned simple baselines would be important too.
- A key motivation for using a probabilistic method such as cDDPM should be to evaluate its skill in emulating the *full distribution* of the response of the atmosphere to climate forcings. Quantitative results only analyze the global mean temperature or other global mean values.

7. The paper lacks critical discussions of broader impacts, ethical considerations, and limitations - notable omissions given its potential applications in solar radiation geoengineering, as mentioned in the abstract. Given the controversial nature of geoengineering and its profound societal implications, these discussions are essential for responsible research in this domain.

Minor:
- The differences between equations 3&4 and 5&6 are trivial and not a good use of space inside the main paper. I'd suggest directly showing 5&6.

References:

[1] The impact of internal variability on benchmarking deep learning climate emulators (https://arxiv.org/abs/2408.05288)

[2] ClimateSet: A large-scale climate model dataset for machine learning; NeurIPS 2023 (https://arxiv.org/abs/2311.03721)

[3] Neural general circulation models for weather and climate; Nature (https://www.nature.com/articles/s41586-024-07744-y)

[4] Neural Networks to Find the Optimal Forcing for Offsetting the Anthropogenic Climate Change Effects; Artificial Intelligence for the Earth Systems (https://doi.org/10.1175/AIES-D-23-0053.1)

**Questions:**

- Section 3.2 *"Classifier-free conditioning"*: Do you mean "Classifier-free guidance"?
- Figure 2: Why are $x_0, ..., x_n$ and $t_0, ..., t_n$ shown as inputs? The input to the network is only *one* of those, right?
- Why do you say that *"Direct conditioning on the forcing field at full resolution (32 × 64) was intractable"*? Have you tried conditioning on it by concatenating it with $x_i$ along the channel dimension before passing these to the U-net as inputs?
- In Fig. 7, what do the forcing fields represent and what is their unit?
- Can you define and explain what the *"linear and nonlinear components of the temperature response as
defined in Lu et al. (2020)*" means?
- Fig. 5 and 7: How is the distribution generated by cDDPM computed? Is it based on an ensemble of generated global mean temperatures (given the same input forcings)? How many ensemble members did you use?

---

### Note · Authors · 2024-11-27

I have read and agree with the venue's withdrawal policy on behalf of myself and my co-authors.